# Trends and outcomes of non-primary PCI at sites without cardiac surgery on-site: The early Michigan experience

Majed Afana[1], Gerald C. Koenig[1,2], Milan Seth[3], Devraj Sukul[3‡], Kathleen M. Frazier[3‡], Sheryl Fielding[3‡], Andrea Jensen[3‡], Hitinder S. Gurm[3,4] *

1 Division of Cardiovascular Medicine, Henry Ford Health System, Detroit, Michigan, United States of America, 2 Wayne State University, School of Medicine, Detroit, Michigan, United States of America, 3 Cardiovascular Medicine, University of Michigan, Ann Arbor, Michigan, United States of America, 4 Cardiovascular Medicine, VA Ann Arbor Healthcare System, Ann Arbor, Michigan, United States of America

☯ These authors contributed equally to this work.
‡ These authors also contributed equally to this work.
* hgurm@med.umich.edu

**Data Availability Statement:** All relevant data necessary to replicate the study findings are within the manuscript and its Supporting Information files. The authors are unable to share the raw data,

## Abstract

### Introduction

Non-primary percutaneous coronary intervention (non-PPCI) recently received certificate of need approval in the state of Michigan at sites without cardiac surgery on-site (cSoS). This requires quality oversight through participation in the BMC2 registry. While previous studies have indicated the safety of this practice, real-world comprehensive outcomes, case volume changes, economic impacts, and readmission rates at diverse healthcare centers with and without cSoS remain poorly understood.

### Methods

Consecutive patients undergoing non-PPCI at 47 hospitals (33 cSoS and 14 non-cSoS) in Michigan from April 2016 to March 2018 were included. Using propensity-matching, patients were analyzed to assess outcomes and trends in non-PPCI performance at sites with and without cSOS.

### Results

Of 61,864 PCI's performed, 50,817 were non-PPCI, with 46,096 (90.7%) performed at sites with cSoS and 4,721 (9.3%) at sites without cSoS. From this cohort, 4,643 propensity-matched patients were analyzed. Rates of major adverse cardiac events (2.6% vs. 2.8%; p = 0.443), in-hospital mortality (0.6% vs. 0.5%; p = 0.465), and several secondary clinical and quality outcomes showed no clinically significant differences. Among a small subset with available post-discharge data, there were no differences in 90-day readmission rates, standardized episode costs, or post-discharge mortality. Overall PCI volume remained stable, with a near three-fold rise in non-PPCI at sites without cSoS.

due to contractual agreements between participating institutions and the BMC2 registry that prohibit data sharing with external agencies. However, the analysis code and metadata to support the study figures is available on request from Annemarie Forrest, Program Manager BMC2 (avassalo@med.umich.edu).

**Funding:** BMC2 registry is supported by the Blue Cross Blue Shield of Michigan (BCBSM) and Blue Care Network as part of the BCBSM Value Partnerships program. The funding source supported data collection and coordination at participating hospitals but had no role in study concept, interpretation of findings, preparation, final approval, or decision to submit the manuscript. No additional specific funding was used for this project or its analysis. Acknowledgements: We are indebted to all the study coordinators, investigators, and patients who participated in the Blue Cross Blue Shield of Michigan Cardiovascular Consortium registry. All authors listed meet the authorship criteria according to the latest guidelines of the International Committee of Medical Journal Editors, and all authors agree with the manuscript. Disclaimer: Although BCBSM and BMC2 work collaboratively, the opinions, beliefs and viewpoints expressed by the authors do not necessarily reflect the opinions, beliefs and viewpoints of BCBSM or any of its employees.

**Competing interests:** Dr. Hitinder S. Gurm receives research funding from BCBSM and the National Institutes of Health and is a consultant for Osprey Medical. This does not alter our adherence to PLOS ONE policies on sharing data and materials. The other authors do not have any direct conflicts or competing interest that could affect the study or its data presentation.

**Abbreviations:** CON, Certificate of need; cSoS, Cardiac surgery on-site; BMC2, Blue Cross Blue Shield of Michigan Cardiovascular Consortium.

## Conclusions

Non-PPCI at centers without cardiac SoS was associated with similar comprehensive outcomes, quality of care, 90-day episode costs, and post-discharge mortality compared with surgical sites. Mandatory quality oversight serves to maintain appropriate equivalent outcomes and may be considered for other programs, including the performance of non-PPCI at ambulatory surgical centers in the near future.

## Introduction

The performance of non-primary percutaneous coronary intervention (non-PPCI), defined as PCI for indications aside from ST-elevation myocardial infarction, cardiac arrest or cardiogenic shock, at sites without cardiac surgery on-site (cSoS) have been under investigation largely through randomized clinical trials [1,2], with the role of surgical presence remaining an ongoing area of interest. The 2011 ACC/AHA/SCAI Guideline for Percutaneous Coronary Intervention upgraded the recommendation for non-primary PCI at facilities without on-site surgical support to Class IIb, as long as appropriate systems of care were developed, and rigorous clinical and angiographic criteria were used for proper patient selection [3]. Subsequent guideline updates have provided further exclusion criteria, based on risk and lesion characteristics, to include more specific institutional, procedural, and provider recommendations regarding the suitability of non-PPCI at centers without cSoS support [4–6]. At the time of the 2014 update, 45 states allowed primary and non-primary PCI without cSoS, 4 allowed only primary PCI without cSoS, and 1 prohibited any PCI without cSoS [6]. Despite the widespread implementation of this practice, guideline recommendations have not been upgraded and remain at a Class IIb.

Previous investigations, especially within the primary PCI literature, have emphasized the point of whether sites without cSoS provide safe and suitable options while maintaining comparable outcomes to surgical sites. As device technologies, procedural techniques, and practice patterns of PCI have evolved, there have been concomitant declines in need for emergency coronary artery bypass surgery after PCI. Further support for the safety of performing non-PPCI at sites without cSoS comes from two previous randomized controlled trials, CPORT-E and MASS COMM, both showing no differences in major adverse cardiovascular events, PCI complications, or need for emergency CABG at centers with and without cSoS [1,2]. A subsequent analysis using the Veterans Affairs (VA) healthcare system, a large integrated healthcare delivery system under quality oversight, confirmed similar in-hospital and 1-year outcomes for generalized outcomes, but also demonstrated improved access as measured by shorter geographic drive times for patients [7].

Despite current guidelines and practice patterns of several other states, the State of Michigan only recently approved the performance of non-PPCI at certificate of need (CON) centers just in March 2016, with the majority of PCI sites having received approval by late 2016. Using the Blue Cross Blue Shield of Michigan Cardiovascular Consortium (BMC2) data, which is a collaborative consortium of multiple, diverse healthcare providers in the State of Michigan implementing quality improvement projects focused on PCI, herein we report the first two years of outcomes and practice patterns for non-PPCI at diverse healthcare sites with and without surgery on-site.

## Methods

### Data source

The Blue Cross Blue Shield of Michigan Cardiovascular Consortium (BMC2) is a prospective, multicenter observational registry that collects demographic, clinical, procedural, and in-hospital outcome data from consecutive PCI cases at all nonfederal hospitals in the state of Michigan. Details of the BMC2 registry, its data collection, and the rigorous and random auditing process have been described previously [8–10]. All data elements have been prospectively defined, and the initial protocol received approval or waiver by the local Institutional Review Board at each institution, as this registry is part of a quality initiative process. No unique patient identifiers were collected and informed consent was waived. All data points were derived from a HIPAA-complaint database. The state of Michigan mandates participation in BMC2 as a requirement for approval for PCI at centers without surgical support. All procedures are submitted for review and the participating hospitals receive extensive quarterly and annual reports detailing procedural and outcome data. Per state of Michigan certificate of need (CON) standards, all sites without on-site surgery are required to have an established plan for the immediate transfer of patients within 60 minutes of travel time from the cardiac catheterization laboratory to a surgical site if necessary [11,12].

"BMC2 registry is supported by the Blue Cross Blue Shield of Michigan (BCBSM) and Blue Care Network as part of the BCBSM Value Partnerships program. The funding source supported data collection and coordination at participating hospitals but had no role in study concept, interpretation of findings, preparation, final approval, or decision to submit the manuscript" [12]. No additional specific funding was used for this project or its analysis. The authors are solely responsible for the design and conduct of this study, the complete analyses, the drafting and editing of the paper, and its final contents.

### Patient and site characteristics

All non-PPCI cases in the BMC2 registry performed at 47 hospitals in Michigan from April 2016 to March 2018 were included. Of these 47 sites, 33 have cSoS, while 14 do not have cSoS. Geographic locations of the participating PCI capable sites superimposed on a population density map of the state of Michigan, using five-year estimates from the 2012 United States American Community Survey obtained from the US Census Bureau, is shown in Fig 1 [13]. We define non-PPCI as PCI performed for any indication excluding ST-elevation myocardial infarction (STEMI), or patients presenting with cardiac arrest or cardiogenic shock. Demographic, clinical and peri-procedural variables were collected and compared.

### Outcomes

The primary composite endpoint was a composite of major adverse cardiovascular events including all-cause in-hospital mortality, contrast-induced nephropathy (CIN), stroke, National Cardiovascular Data Registry (NCDR)–defined bleeding, and major bleeding defined as ≥5 g/dL drop in hemoglobin. A secondary post hoc composite endpoint of in-hospital mortality, cerebrovascular accident(CVA)/stroke, stent thrombosis, and target lesion revascularization was also assessed. Secondary outcomes were independent measures of all-cause in-hospital mortality, major bleeding, RBC/whole blood transfusion, other vascular complications requiring transfusion, CVA/stroke, cardiogenic shock, heart failure, subacute stent thrombosis, target lesion revascularization, new requirement for dialysis, urgent/emergent CABG, CIN, and length of stay. CIN was defined as an increase in serum creatinine ≥0.5 mg/dL from pre- to post-PCI measurement [14]. Vascular complications included a composite of

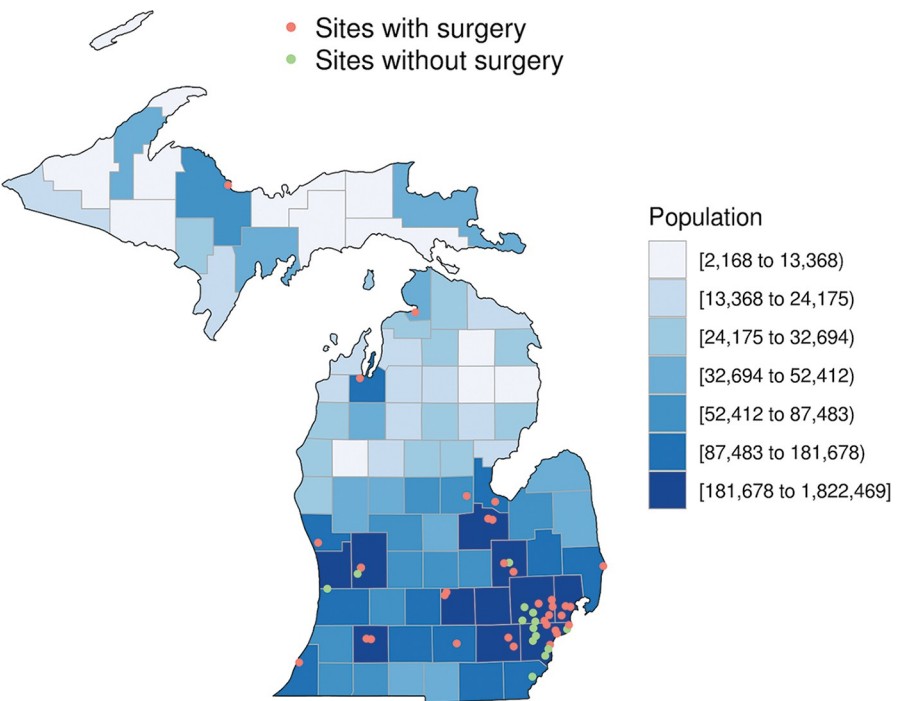

**Fig 1. Participating sites in study.** Locations of participating PCI capable sites included in this study superimposed on a county-based population density map of the state of Michigan. Data on population density derived from the 2012 United States American Community Survey five-year estimates obtained from the US Census Bureau.

pseudoaneurysm, arteriovenous fistula, access site occlusions, dissections, peripheral embolizations, or other complications requiring intervention, including surgical repair, thrombin injection, or angioplasty. Bleeding events, as defined by the NCDR *CathPCI Registry*, included suspected or confirmed bleeding observed and documented in the medical record that was associated with any of the following: (1) hemoglobin drop >3 g/dL, (2) transfusion of whole blood or packed red blood cells, and (3) procedural intervention or surgery at the bleeding site to stop bleeding.

In addition to clinical outcomes, we sought to assess the trends in both total and non-PPCI volumes and distribution among sites with and without cSoS during the study period on a quarterly basis.

### Readmission rates, costs, and long-term mortality

Among a small subset of patients that could be linked to administrative Medicare claims, we performed additional analyses comparing post-discharge readmission rates, standardized 90-day episode costs, and long-term mortality. Medicare data was made available through coordination with the Michigan Value Collaborative (MVC), which constructs 90-day episodes of care using Medicare administrative data constructed from Commercial BCBSM PPO, Blue Care Network, Medicare Advantage PPO and HMO, as well as Medicare Fee-For-Service claims data [15]. Using unique matching of multiple indirect patient and procedural identifiers, including hospital and operator National Provider Identifier numbers, admission, discharge and procedural dates for the index hospitalization, patient gender and date of birth, with PCI episode data from our collaboration with the MVC, successful linking of data was performed to determine post-discharge outcomes as above. Standardized episode cost was an

adjusted cost to reflect a common Medicare charge master so that differences in payer, and differences in hospital contracts are not reflected in the comparison. Post-discharge mortality at 90 days was obtained from the Medicare beneficiary file, which includes the date of death for deceased beneficiaries.

### Statistical analysis

Baseline characteristics between sites with and without surgery were compared using Pearson $\chi^2$ testing for categorical variables and Wilcoxon rank-sum and Student $t$ tests for continuous variables. Absolute standardized differences (ASD) were estimated, and a 10% threshold was used as an indicator of clinically meaningful imbalances. Continuous variables were summarized using mean ± SD. Data from all PCI cases performed in the State of Michigan from April 2016 to March 2018 were collected. From this, we excluded patients presenting for PCI for STEMI, cardiogenic shock or cardiac arrest to create the overall non-PPCI cohort. Demographic, peri-procedural and outcomes data were then defined and presented. In order to assess outcomes of comparable patients, we performed 1:1 propensity score based greedy matching using logistic regression models adjusting for 21 baseline clinical and demographic variables (S1 Table). A caliper on the propensity score was used, requiring that matched pairs have propensity score values within 0.25 standard deviations of each other. Lesion categories were assumed to be binomial, given the possibility for cases to have multiple locations identified; therefore, $P$ values for each location were provided. In addition to the nominal p-value, an adjusted p-value accounting for multiple comparisons using the Bonferroni method was calculated for our outcomes.

Further subgroup analyses of the non-PPCI cohort was performed after exclusion of high-risk procedures (procedures that might be preferentially performed at institutions with surgical back up), defined as interventions on the left main coronary artery, chronic total occlusions, bifurcation lesions, patients with a pre-procedural ejection fraction less than 30%, or the use of atherectomy devices, with repeat analysis of baseline characteristics and outcomes. Finally, although limited in number due to imposed restrictions, we assessed outcomes of the high-risk patients alone between site types. All analyses were performed using R statistical software version 3.3 [16].

## Results

### Patient and procedural characteristics

Between April 1, 2016, and March 31, 2018, a total of 61,864 PCIs were performed in the state of Michigan with 55,247 (89.3%) performed at sites with cSoS and 6,617 (10.7%) at sites without cSoS. From this overall cohort, a total of 50,817 cases of non-PPCI existed, with 46,096 (90.7%) at sites with cSoS and 4,721 (9.3%) at sites without cSoS. There was a progressive rise in total case volume at sites without cSoS from 6.7% of overall statewide PCIs in the first quarter of the study period to 13.9% in the last quarter, an approximately two-fold increase, with a concomitant decline in cases at sites with cSoS. The overall PCI volume, however, remained relatively stable with a proportional shift away from sites with cSoS to sites without cSoS (Fig 2A). When looking specifically at the non-PPCI cohort, the same pattern held true, however, there was a near three-fold rise in non-PPCI at non-cSoS (4.6% in the first quarter of the study year period to 13.0% at the end of study period) as shown in Fig 2B. Furthermore, among non-cSoS, there appeared to be heterogeneity in the ramp-up of PCI volume when looking at the first 12-month period for each site after beginning to perform non-PPCIs (S1 Fig). Total PCI volume at the individual 14 non-surgical sites over the 2-year study period is shown in S2 Fig.

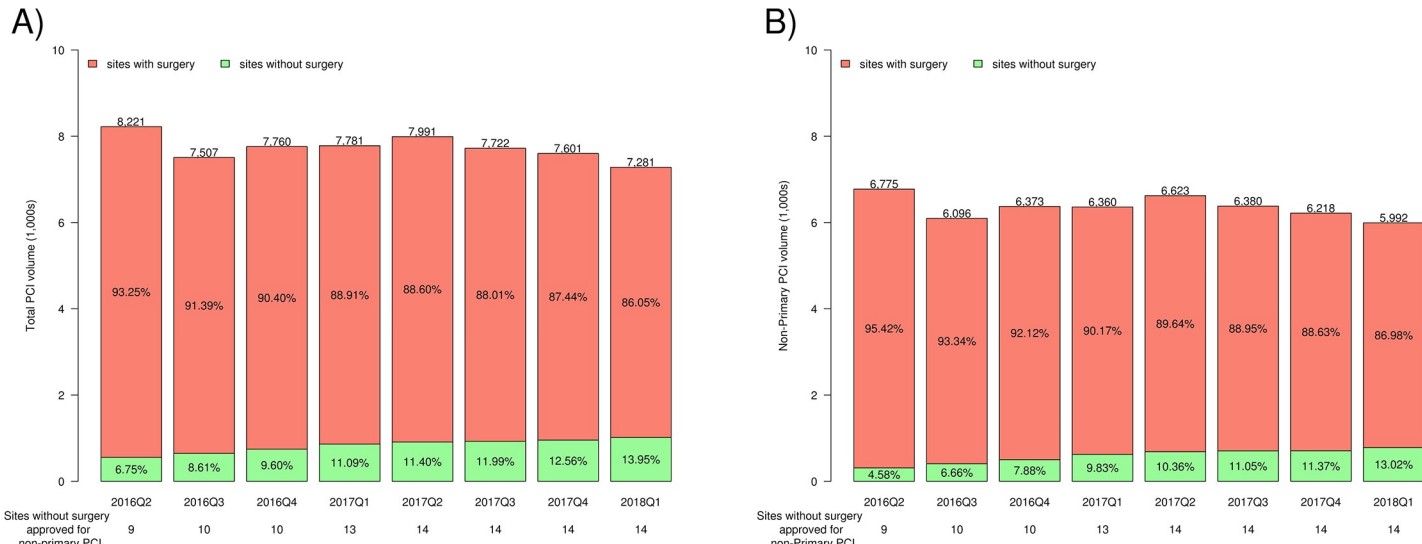

**Fig 2. Trends of PCI performed in Michigan during study period.** Trends and site-specific distribution of (A) all PCIs performed in Michigan, and (B) non-primary PCI cases, during the study period as demonstrated in a quarterly fashion. Number of participating sites without cardiac surgery on-site per quarter displayed below each quarter.

Complete baseline characteristics and clinical variables of the overall unmatched non-PPCI cohorts can be found in S2 Table. Compared with sites without cSoS, sites with cSoS tended to have older patients (66.7 ± 11.5 vs. 64.9 ± 11.4 years; p < 0.001) with higher rates of dyslipidemia (84.3% vs. 77.4%; p < 0.001), peripheral arterial disease (16.4% vs. 11.4%; p < 0.001), prior PCI (51.4% vs. 43.9%; p < 0.001), CABG (20.0% vs. 11.1%; p < 0.001), and heart failure (21.9% vs. 16.3%; p < 0.001). Patients were more likely to present with non-ST elevation myocardial infarction at sites without cSoS (34.7% vs. 28.4%; p < 0.001) compared with sites with cSoS. Surgical sites tended to do more PCI to left main coronary arteries (4.0% vs. 1.0%; p < 0.001), bypass grafts (6.4% vs. 3.5%; p < 0.001), and chronic total occlusions (4.8% vs. 1.9%; p < 0.001). At this time, atherectomy device usage is not permitted at sites without cSoS under the Michigan CON regulations. There were no major differences seen in stent types used.

Our propensity-matching model successfully constructed matched cohorts of 4,643 cases at each of the sites with baseline characteristics and clinical variables displayed in Table 1. Using an absolute standardized difference threshold of >10% for clinical significance, notable differences remained in patients with history of CABG (17.8% vs. 11.3%; p < 0.001) and dyslipidemia (82.2% vs. 77.4%; p < 0.001), and the performance of PCI on left main coronary arteries (3.3% vs. 1.1%; p < 0.0001), chronic total occlusions (4.9% vs. 1.9%; p < 0.0001), along with 2- and 3-vessel PCI. Additional differences in arterial access site usage, intra-procedural medications, and secondary oral anti-platelet medications are also shown in Table 1.

## Outcome analysis

Information regarding outcomes in the overall unmatched non-PPCI cohort is shown in S3 Table. In brief, there were only minor statistically significant differences in secondary outcomes, for instance, blood transfusions, heart failure, and increased length of stay were more likely at sites with cSoS compared to those without cSoS, but no clinically relevant differences were seen. Outcomes of the overall matched non-PPCI cohort are shown in Table 2. There was no statistically significant difference in the primary composite outcome of major adverse

**Table 1. Baseline characteristics of propensity-matched cohorts of elective PCI's performed during study period.**

| | Sites with Surgery | %cases | Sites Without Surgery | %cases | P-value | ASD (%) |
|---|---|---|---|---|---|---|
| N | 4,643 | | 4,642 | | | |
| *Demographics* | | | | | | |
| Age, years | 65.3 ± 11.9 | | 65.0 ± 11.5 | | p = 0.184 | 2.75 |
| Male | 3,145 | 67.7% | 3,023 | 65.1% | p = 0.008 | 5.54 |
| White | 3,909 | 84.2% | 4,066 | 87.6% | p < 0.001 | 9.78 |
| *Clinical History* | | | | | | |
| Hypertension | 4,087 | 88.0% | 3,960 | 85.3% | p < 0.001 | 8.01 |
| Dyslipidemia | 3,816 | 82.2% | 3,592 | 77.4% | p < 0.001 | 12.09 |
| Diabetes Mellitus | 1,917 | 41.3% | 1,899 | 40.9% | p = 0.704 | 0.79 |
| Current/Recent Smoker (<1 year) | 1,221 | 26.3% | 1,297 | 27.9% | p = 0.076 | 3.68 |
| Family History of Premature CAD | 572 | 12.3% | 678 | 14.6% | p = 0.001 | 6.70 |
| Peripheral Arterial Disease | 518 | 11.2% | 535 | 11.5% | p = 0.575 | 1.16 |
| Prior Myocardial Infarction | 1,674 | 36.1% | 1,561 | 33.6% | p = 0.014 | 5.10 |
| Prior PCI | 2,261 | 48.7% | 2,029 | 43.8% | p < 0.001 | 9.90 |
| Prior CABG | 824 | 17.8% | 522 | 11.3% | p < 0.001 | 18.53 |
| Prior Heart Failure | 915 | 19.7% | 764 | 16.5% | p < 0.001 | 8.45 |
| Heart Failure within 2 weeks | 625 | 13.5% | 621 | 13.4% | p = 0.922 | 0.20 |
| Prior Valve Surgery/Procedure | 100 | 2.2% | 80 | 1.7% | p = 0.134 | 3.11 |
| Chronic Lung Disease | 821 | 17.7% | 810 | 17.5% | p = 0.768 | 0.61 |
| Currently on Dialysis | 136 | 2.9% | 135 | 2.9% | p = 0.948 | 0.14 |
| Cerebrovascular Disease | 733 | 15.8% | 589 | 12.7% | p < 0.001 | 8.89 |
| GFR, mL/min/1.73m$^2$ (CKD-EPI) | 74.2 ± 24.6 | | 75.4 ± 24.1 | | p = 0.016 | 5.00 |
| Body Mass Index, kg/m$^2$ | 31.2 ± 9.9 | | 30.9 ± 6.7 | | p = 0.150 | 2.99 |
| *CAD Presentation* | | | | | | |
| NSTEMI | 1,618 | 34.8% | 1,620 | 34.9% | p = 0.959 | 0.11 |
| Unstable Angina | 2,061 | 44.4% | 2,113 | 45.5% | p = 0.274 | 2.27 |
| Stable Angina | 642 | 13.8% | 557 | 12.0% | p = 0.009 | 5.45 |
| Symptoms unlikely to be ischemic | 142 | 3.1% | 159 | 3.4% | p = 0.318 | 2.07 |
| No symptoms, no angina | 180 | 3.9% | 193 | 4.2% | p = 0.491 | 1.43 |
| *Access Site* | | | | | | |
| Femoral | 2,424 | 52.2% | 1,994 | 43.0% | p < 0.001 | 18.57 |
| Radial | 2,203 | 47.4% | 2,631 | 56.7% | p < 0.001 | 18.61 |
| Brachial | 7 | 0.2% | 7 | 0.2% | p = 0.999 | 0.00 |
| Other | 9 | 0.2% | 8 | 0.2% | p = 0.809 | 0.50 |
| *Peri-Procedural Variables & Complications* | | | | | | |
| IABP | 27 | 0.6% | 29 | 0.6% | p = 0.785 | 0.56 |
| Perforation | 16 | 0.3% | 16 | 0.3% | p = 0.998 | 0.01 |
| Significant Dissection | 29 | 0.6% | 26 | 0.6% | p = 0.687 | 0.84 |
| Pre-PCI LVEF, mean % + SD | 52.5 ± 12.9 | | 52.8 ± 12.0 | | p = 0.258 | 2.66 |
| Contrast Volume, mean mL + SD | 157.1 ± 66.4 | | 161.1 ± 66.8 | | p = 0.004 | 5.97 |
| *Vessel(s) Intervened Upon* | | | | | | |
| All Left Main | 154 | 3.3% | 49 | 1.1% | p < 0.001 | 15.51 |
| Left Anterior Descending | 2072 | 44.6% | 2019 | 43.5% | p = 0.277 | 2.28 |
| Left Circumflex | 1366 | 29.4% | 1256 | 27.1% | p = 0.019 | 5.25 |
| Right Coronary Artery | 1534 | 33.0% | 1536 | 33.1% | p = 0.965 | 0.11 |
| Bypass Graft | 249 | 5.4% | 162 | 3.5% | p < 0.001 | 9.12 |
| Chronic Total Occlusion | 228 | 4.9% | 88 | 1.9% | p < 0.001 | 16.69 |

*(Continued)*

**Table 1.** (Continued)

|  | Sites with Surgery | %cases | Sites Without Surgery | %cases | P-value | ASD (%) |
|---|---|---|---|---|---|---|
| Bifurcation | 368 | 7.9% | 375 | 8.1% | p = 0.789 | 0.56 |
| Lesion Data Missing | 9 | 0.2% | 106 | 2.3% | p < 0.001 | 18.98 |
| *# Vessels Intervened Upon* |  |  |  |  |  |  |
| 1 vessel PCI | 3968 | 85.5% | 4115 | 88.6% | p < 0.001 | 9.50 |
| 2 vessel PCI | 600 | 12.9% | 410 | 8.8% | p < 0.001 | 13.17 |
| 3 vessel PCI | 66 | 1.4% | 11 | 0.2% | p < 0.001 | 13.09 |
| *Device Used* |  |  |  |  |  |  |
| Bare Metal Stent only | 180 | 3.9% | 250 | 5.4% | p < 0.001 | 7.18 |
| Drug-Eluting Stent only | 4020 | 86.6% | 4031 | 86.8% | p = 0.737 | 0.75 |
| Balloon only | 324 | 7.0% | 264 | 5.7% | p = 0.012 | 5.30 |
| BMS + DES | 19 | 7.0% | 8 | 0.2% | p = 0.052 | 4.40 |
| Device Data Missing | 100 | 2.2% | 89 | 1.9% | p = 0.463 | 1.67 |
| Atherectomy device | 142 | 3.1% | 0 | 0.0% | p < 0.001 | 25.12 |
| *Intra-procedural Medications* |  |  |  |  |  |  |
| IV UFH | 4342 | 93.5% | 4486 | 96.7% | p < 0.001 | 14.47 |
| Bivalirudin | 659 | 14.2% | 433 | 9.3% | p < 0.001 | 15.50 |
| Bivalirudin + GPI | 18 | 0.4% | 23 | 0.5% | p = 0.440 | 1.63 |
| GPI + UFH | 800 | 17.2% | 1110 | 23.9% | p < 0.001 | 16.59 |
| *Oral Antiplatelets Used* |  |  |  |  |  |  |
| Aspirin | 4569 | 98.4% | 4251 | 91.6% | p < 0.001 | 31.70 |
| Clopidogrel | 2510 | 54.1% | 2126 | 45.8% | p < 0.001 | 16.58 |
| Prasugrel | 399 | 8.6% | 215 | 4.6% | p < 0.001 | 15.99 |
| Ticagrelor | 1691 | 36.4% | 2015 | 43.4% | p < 0.001 | 14.30 |

ASD = absolute standardized difference; BMS = bare metal stent; CAD = coronary artery disease; CABG = coronary artery bypass graft; CKD-EPI = Chronic Kidney Disease Epidemiology Collaboration; DES = drug eluting stent; GFR = glomerular filtration rate; GPI = glycoprotein inhibitor; IABP = intra-aortic balloon pump; NSTEMI = non-ST elevation myocardial infarction; UFH = unfractionated heparin.

cardiovascular events (2.6% vs. 2.8%; p = 0.443), secondary composite outcome of MACE (1.2% vs. 0.8%; p = 0.060), or all-cause in-hospital mortality (0.6% vs. 0.5%; p = 0.465) at sites with and without cSoS. Furthermore, there were also no statistically significant differences in secondary outcomes of major bleeding, RBC/whole blood transfusion, other vascular complications requiring transfusion, subacute stent thrombosis, target lesion revascularization, new requirement for dialysis, urgent/emergent CABG, CIN, or length of stay. Rates of CVA/stroke and heart failure were lower at non-surgical sites, but did not meet criteria for clinical significance. This was further displayed after adjustment accounting for multiple comparisons using the Bonferroni method for CVA (0.4% vs. 0.1%; p = 0.126), while heart failure still maintained statistical significance without clinical significance. (1.9% vs. 1.1%; p = 0.014). After exclusion of high-risk cases from the matched cohort, we again found no significant differences between the two groups (S4 Table). Lastly, the comparison of outcomes in matched high-risk patients, an overall low volume of cases given CON Restrictions at non-cSOS, showed no differences in major or secondary outcomes between the two sites. (S5 Table).

## Readmission rates, costs, and long-term mortality

Of the 50,817 non-PPCI cases in the analysis cohort, 10,104 (19.9%) Medicare fee-for-service patients were matched to MVC claims-based episodes. Of the matched cases, 9,340 (20.3%)

**Table 2. Outcomes and complications of matched elective PCI cohorts at sites with and without surgery on-site.**

| | Sites with Surgery | %cases | Sites Without Surgery | %cases | P-value (nominal) | P-value (Bonferroni adjusted) | ASD (%) |
|---|---|---|---|---|---|---|---|
| *N* | 4,643 | | 4,642 | | | | |
| Primary Composite Endpoint | 119 | 2.6% | 131 | 2.8% | p = 0.443 | p = 1.000 | 1.60 |
| Secondary Composite Endpoint | 56 | 1.2% | 37 | 0.8% | p = 0.060 | p = 0.902 | 4.11 |
| In-Hospital Mortality | 26 | 0.6% | 21 | 0.5% | p = 0.465 | p = 1.000 | 1.52 |
| Major Bleeding | 19 | 0.5% | 14 | 0.4% | p = 0.728 | p = 1.000 | 1.10 |
| RBC/Whole Blood Transfusion | 61 | 1.3% | 53 | 1.1% | p = 0.452 | p = 1.000 | 1.56 |
| Other Vascular Complications Requiring Transfusion | 14 | 0.3% | 7 | 0.2% | p = 0.126 | p = 1.000 | 3.17 |
| CVA/Stroke | 19 | 0.4% | 6 | 0.1% | p = 0.009 | p = 0.126 | 5.41 |
| Cardiogenic Shock | 37 | 0.8% | 41 | 0.9% | p = 0.647 | p = 1.000 | 0.95 |
| Heart Failure | 87 | 1.9% | 49 | 1.1% | p = 0.001 | p = 0.014 | 6.81 |
| Subacute stent thrombosis | 5 | 0.1% | 6 | 0.1% | p = 0.774 | p = 1.000 | 0.63 |
| Target lesion revascularization | 15 | 0.3% | 8 | 0.2% | p = 0.210 | p = 1.000 | 3.03 |
| CABG (urgent/emergent status) | 22 | 0.5% | 15 | 0.3% | p = 0.250 | p = 1.000 | 2.39 |
| Contrast-Induced Nephropathy | 72 | 1.8% | 64 | 1.9% | p = 0.795 | p = 1.000 | 0.61 |
| New Requirement for Dialysis | 6 | 0.1% | 5 | 0.1% | p = 0.763 | p = 1.000 | 0.62 |
| Length of Stay (days) | 2.8 ± 5.6 | | 2.6 ± 3.0 | | p = 0.117 | p = 1.000 | 3.25 |

ASD = absolute standardized difference; CABG = coronary artery bypass graft; CVA = cerebrovascular accident; MACE = major adverse cardiovascular event; PCI = percutaneous coronary intervention; RBC = red blood cell.

were from sites with cSoS and 764 (16.2%) were from sites without cSoS. The 90-day readmission rates (18.8% vs. 20.0%; p = 0.400) and standardized episode costs ($26,457.25 vs. $26,279.80; p = 0.902) were similar at sites with and without cSoS, respectively. Time to post-discharge mortality was also similar (p = 0.836) and displayed in Fig 3.

## Subgroup analysis

In a sensitivity analysis, we subsequently removed cases deemed to be high-risk for sites without cSoS to analyze a more representative matched cohort of 3,967 cases intended to be allocated to CON centers during the introductory roll out phase. The baseline characteristics of this cohort are shown in S6 Table. Differences in history of prior CABG and access site usage held true even after excluding high-risk cases. Within both the unmatched and matched cohorts, contrast volume was not significantly different, but after exclusion of high-risk cases, there was evidence of slightly higher mean contrast volume at sites without cSoS (S4 and S6 Tables), with no increase in CIN rates or new requirement for dialysis.

## Discussion

This study shows that non-PPCI at multiple, diverse healthcare sites without cSoS can be safely performed with quality oversight with no significant differences in comprehensive outcomes, including in-hospital MACE, mortality and complication rates along with 90-day readmission rates, costs, and long-term mortality, when compared with sites with cSoS. Importantly, there were no differences in the need for urgent/emergent CABG, CIN, length of stay, or bleeding complications, which have been closely linked to patient outcome measures through multiple studies. Despite differences in vascular access sites and variability in intra-procedural anticoagulant and antiplatelet use, these outcomes were maintained. Furthermore, even after exclusion

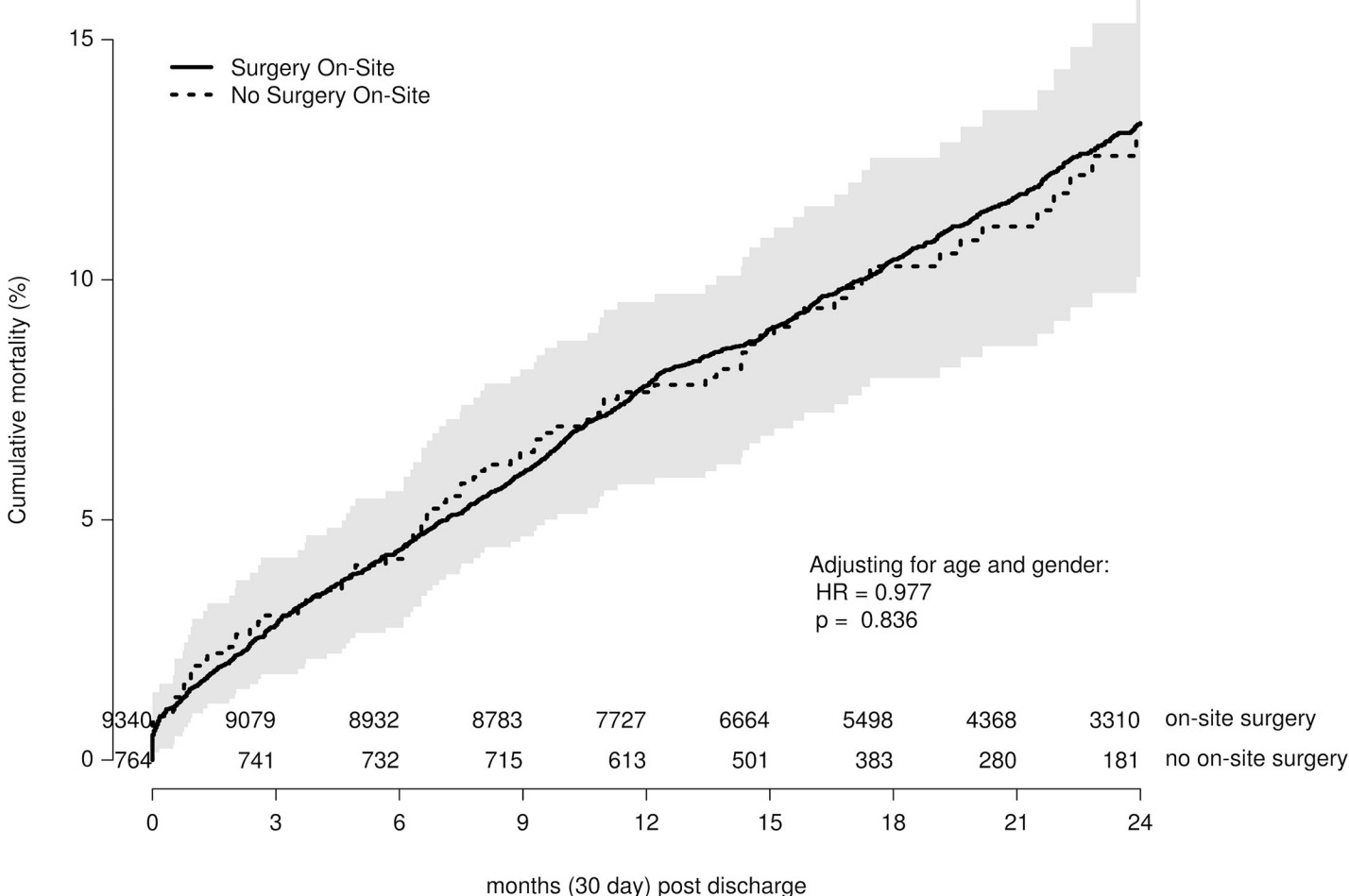

**Fig 3. Long-term mortality of patient at sites without surgery on-site.** Kaplan-Meier curve of long-term mortality among subset of matched Medicare patients at sites with (solid line) and without (dotted line) surgery on-site. Curve demonstrates no significant difference in overall time to post-discharge mortality. The solid gray bars represent the 95% confidence interval around the estimates of cumulative mortality.

of higher risk patients or lesion categories, no clinically significant differences in outcomes remained.

Our study also interestingly found that despite approval of 14 additional sites without cSoS to perform non-PPCI during the study period, an increase of over 40% for the State, the overall volume of PCI procedures performed remained stable. This was the result of a clear shift in case volume towards non-surgical sites in accommodating patients to remain closer to their residence and family, and within the same integrated or affiliated healthcare system. These patient-centered accommodations further showed no apparent detriment to the episode costs, a concern raised prior to the CON acceptance by third party payers.

Our findings are consistent with previous selective studies showing the safety of non-PPCI at sites without on-site surgery [1,2]. Specifically, we found no statistically or clinically significant differences in the rates of urgent/emergent CABG at sites with and without cSoS, emphasizing the safety of performing non-PPCI at non-surgical sites, as well as confirming the well-established low rates of urgent/emergent CABG secondary to PCI complications during this contemporary period. Two prior randomized controlled trials, the CPORT-E and MASS COMM trials, looking at non-PPCI patients in a selective and equipoise manner only showed

no differences in major adverse cardiovascular events, PCI complications, or need for emergency CABG at centers with and without on-site surgery. CPORT-E, published in 2012, showed both 6-week mortality rates and 9-month rates of MACE to be non-inferior to hospitals with cSoS. Of note, only sites performing >200 PCI/year, and operators performing >75 PCI/year were included in that study [1]. The MASS COMM trial, published in 2013, used a similar study design and found 30-day and 12-month rates of MACE to also be non-inferior at hospitals without cSoS to sites with cSoS [2]. These findings have been reproduced in a retrospective analysis from the United Kingdom, where the majority of PCI facilities do not have cSoS, using a large non-PPCI cohort of 99,438 patients at centers without surgery versus 195,316 patients at centers with on-site surgery and showed no significant difference in mortality at 30 days, 1 year, and 5 years [17]. Observational data using the VA health care system clinical reporting and tracking analyzed sites with and without cSoS receiving PCI for all indications, including non-PPCI, demonstrated no differences in emergent CABG, 1-year adjusted mortality and myocardial infarction rates, while also increasing geographic access to VA patients [7]. Additional meta-analyses, published before CPORT-E and MASS COMM trials, also showed no increased mortality, need for emergency bypass, or PCI complications for non-PPCI at centers with and without surgical support [18,19]. Our study's ability to reproduce and extend those of prior studies with regards to the safety of performance of non-PPCI within the state of Michigan are strengthened by the use of a rigorously monitored statewide quality improvement initiative. Furthermore, unlike randomized controlled trials, which are highly regulated with regards to treatment approaches, and meta-analyses that are limited by variable sample sizes, selection of included studies, and heterogeneity of methodology and data analysis, our study is reflective of real-world practices.

In addition to confirming comparable outcomes of non-PPCI at non-surgical sites, we have uniquely shown, among a small subset of Medicare patients, that 90-day readmission rates, standardized 90-day episode costs and long-term mortality are similar. This vital information lends further support to not only the clinical safety, but also the economic feasibility and practicality of these practices. As health care costs continue to escalate with increasing prevalence of cardiovascular disease and constant advancement of medical technologies and equipment, the economic evaluations of health care delivery are essential. Previous studies have shown that a significant portion of post-PCI costs are related to hospital readmissions, lengths of stay, and post-procedural complications or post-acute care [20–22]. Using large readmission and administrative claim databases, analyzing hundreds of thousands of patients undergoing PCI, these studies have shown clear associations between longer lengths of stay and post-procedural complication rates with increased costs. These factors are essential to the consideration of expanding non-PPCI practices to non-surgical sites, and this study's demonstration of comparable outcomes, low complication rates, lengths of stay, and readmission rates reinforce that point.

The acceptance of the centers without cSoS raised concerns of a detrimental influence on procedural volumes and appropriate use in the process of meeting CON requirements for the individual centers. During the study period analyzed, there was a clear progressive rise in PCI volume at non-surgical sites that was matched by a contemporaneous decline at surgical sites. The analysis consisted of both the added PCI volume from sites entering at variable times during the observation period, and the overall increase in volume at the given sites without cSoS. This resulted in an overall annual PCI volume that remained stable, suggesting that addition of PCI sites resulted in more redistribution with greater access within a given healthcare system, and did not result in significant increases in overall volumes. Current trends are also showing a migration of cardiac procedures to ambulatory surgery centers (ASCs), initially involving pacemaker implants followed by diagnostic cardiac catheterizations. Newly accepted Centers

for Medicare & Medicaid Services (CMS) rules have cleared the way for PCI to be conducted in these ASCs in 2020, directly competing with both centers having cardiac surgery on-site and those without cSoS. Both safety and economic factors have been a concern raised with this new proposal of care. Using the analysis and findings of our study involving the State of Michigan's CON process with specific regulations, guidelines and quality assurance measures providing not only comprehensive comparable outcomes but also economic feasibility, may serve as a means to lead the way towards guidance in the acceptance and progression to ASCs performing PCI.

Despite several strengths of this study, there are important limitations. While the analysis uses observational, nonrandomized data from a single-state registry, we used propensity score matching to present well-balanced comparisons between sites. The BMC2 database only includes in-hospital data, therefore, long-term outcomes, including post-procedural mortality and complications, could not be fully investigated. However, as shown above, the equipoise in in-hospital outcome data, and the limited post-discharge mortality rates, are congruent to extend to additional outcomes, and are also consistent with other limited studies that have already shown these to be similar between surgical and non-surgical sites for non-PPCI. Given variability in the rates of pre- and post-procedural biomarker assessment among sites, we did not include post-procedural myocardial infarction as an outcome of interest. While differences remained in pharmacotherapy and treatment strategies after matching, adjustments were made on patient level variables and not treatment level variables to understand differences in treatment. While this may reflect differences in treatment strategies, including cost-conscious practice patterns, institutional or operator preferences, we are unable to fully explain why these differences exist [12]. The post-discharge data on long-term mortality, readmission rates, and standardized episode costs is limited by the small subset of patients available for matching from the MVC claims-based episodes and would benefit from additional study. Finally, while the sub-group of high-risk patients, such as left main coronary artery disease, bifurcation lesions, requirement for atherectomy, and low ejection fraction was small, we still showed a signal of no difference in outcomes, which may lend credence to the extension of these practices to all PCI facilities. This subgroup ultimately represents the next stage of system approval with the ultimate expansion of these practices to all PCI facilities. However, further studies will be needed to confirm the safety of these procedures with larger patient populations.

## Conclusion

Under Michigan's recently adopted CON requirement permitting non-primary PCI at multiple, diverse healthcare facilities without surgery on-site, there are similar comprehensive outcomes, complications and readmission rates and costs compared to facilities with surgery on-site. These findings reinforce the clinical and economic safety of performing non-primary PCI at either site. The use of a robust system of data collection, auditing and quality improvement may provide a template for ensuring safe and optimal expansion of PCI services across centers with no cardiac surgery on site, including the newest domain of ambulatory surgery centers.

## Supporting information

**S1 Table. Baseline variables included in the propensity-matched model.**
(DOCX)

**S2 Table. Baseline characteristics of overall unmatched non-primary PCI cohort during study period.**
(DOCX)

**S3 Table. Overall clinical and procedural outcomes, and major complications of unmatched non-primary cohorts at sites with and without on-site surgery.**
(DOCX)

**S4 Table. Clinical and procedural outcomes, and major complications at sites with and without on-site surgery excluding high-risk patients.**
(DOCX)

**S5 Table. Clinical and procedural outcomes, and major complications at sites with and without on-site surgery of high-risk patient subset.**
(DOCX)

**S6 Table. Baseline characteristics of propensity-matched cohorts of non-primary PCI's not including high-risk cases at sites with and without cardiac surgery.**
(DOCX)

**S1 Fig. PCI volume trend during first 12-month period post-approval.** Non-primary PCI volume from time of approval through first 12-month period post-approval for (a) all 14 participating sites individually, and (b) combined PCI volume for all sites during each of the months. Notable heterogeneity seen between sites, with progressive ramp-up of volume during the first 12 month period for each site post-initiation of non-primary PCI practices.
(TIF)

**S2 Fig. Total PCI volumes at individual non-surgical sites during study period.** Total PCI volumes over the 2-year study period for each of the 14 non-surgical sites included in the study are shown, with breakdown of both primary and non-primary PCI.
(TIFF)

## Acknowledgments

We are indebted to all the study coordinators, investigators, and patients who participated in the Blue Cross Blue Shield of Michigan Cardiovascular Consortium registry. All authors listed meet the authorship criteria according to the latest guidelines of the International Committee of Medical Journal Editors, and all authors agree with the manuscript.

## Disclaimer

Although BCBSM and BMC2 work collaboratively, the opinions, beliefs and viewpoints expressed by the authors do not necessarily reflect the opinions, beliefs and viewpoints of BCBSM or any of its employees.

## Author Contributions

**Conceptualization:** Majed Afana, Milan Seth, Hitinder S. Gurm.

**Formal analysis:** Majed Afana, Milan Seth.

**Funding acquisition:** Hitinder S. Gurm.

**Methodology:** Majed Afana, Milan Seth, Hitinder S. Gurm.

**Supervision:** Hitinder S. Gurm.

**Validation:** Milan Seth, Devraj Sukul.

**Writing – original draft:** Majed Afana.

**Writing – review & editing:** Gerald C. Koenig, Devraj Sukul, Kathleen M. Frazier, Sheryl Fielding, Andrea Jensen, Hitinder S. Gurm.

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
