## [Decision Letter · Decision Letter 0]

22 Jun 2020

PONE-D-20-15844

Trends and Outcomes of Elective PCI at Sites Without Cardiac Surgery On-Site: The Early Michigan Experience

PLOS ONE

Dear Dr. Gurm,

Thank you for submitting your manuscript to PLOS ONE. After careful consideration, we feel that it has merit but does not fully meet PLOS ONE’s publication criteria as it currently stands. Therefore, we invite you to submit a revised version of the manuscript that addresses the points raised during the review process.

Reviewers have raised concerns about issues of clinical practice and about the actual definition of elective PCI. Moreover, it is unusual to include CIN in the composite endpoint of MACE, although one has to acknowledge that the Authors have extensively worked on such topic and have a huge amount of data. This point, along with center volume of PCI, should be properly discussed as suggested.

We look forward to receiving your revised manuscript.

Kind regards,

Giuseppe Andò, M.D., Ph.D.

Academic Editor

PLOS ONE

Journal Requirements:

2. Please refer to the specific statistical analyses performed as well as any post-hoc corrections to correct for multiple comparisons. If these were not performed please justify the reasons.

Please refer to our statistical reporting guidelines for assistance (https://journals.plos.org/plosone/s/submission-guidelines.#loc-statistical-reporting)

3. In the ethics statement in the manuscript and in the online submission form, please provide additional information about the patient records used in your retrospective study.

Specifically, please ensure that you have discussed whether all data were fully anonymized before you accessed them and/or whether the IRB or ethics committee waived the requirement for informed consent.

If patients provided informed written consent to have data from their medical records used in research, please include this information.

4.We noticed you have some minor occurrence of overlapping text with the following previous publication(s), which needs to be addressed:

-https://www.sciencedirect.com/science/article/abs/pii/S0002870317303162?via%3Dihub

In your revision ensure you cite all your sources (including your own works), and quote or rephrase any duplicated text outside the methods section. Further consideration is dependent on these concerns being addressed.

5.We note that you have indicated that data from this study are available upon request. PLOS only allows data to be available upon request if there are legal or ethical restrictions on sharing data publicly. For information on unacceptable data access restrictions, please see http://journals.plos.org/plosone/s/data-availability#loc-unacceptable-data-access-restrictions.

6. Thank you for stating the following in the Competing Interests section:

'Competing interests: Hitinder S. Gurm receives research funding from BCBSM and the National Institutes of Health and is a consultant for Osprey Medical. The authors have declared that no competing interests exist.'

a. Please confirm that this does not alter your adherence to all PLOS ONE policies on sharing data and materials, by including the following statement: "This does not alter our adherence to  PLOS ONE policies on sharing data and materials.” (as detailed online in our guide for authors http://journals.plos.org/plosone/s/competing-interests).  If there are restrictions on sharing of data and/or materials, please state these.

Please note that we cannot proceed with consideration of your article until this information has been declared.

7. PLOS requires an ORCID iD for the corresponding author in Editorial Manager on papers submitted after December 6th, 2016. Please ensure that you have an ORCID iD and that it is validated in Editorial Manager. To do this, go to ‘Update my Information’ (in the upper left-hand corner of the main menu), and click on the Fetch/Validate link next to the ORCID field. This will take you to the ORCID site and allow you to create a new iD or authenticate a pre-existing iD in Editorial Manager. Please see the following video for instructions on linking an ORCID iD to your Editorial Manager account: https://www.youtube.com/watch?v=_xcclfuvtxQ

Reviewers' comments:

Reviewer's Responses to Questions

**Comments to the Author**

1. Is the manuscript technically sound, and do the data support the conclusions?

Reviewer #1: Partly

Reviewer #2: Yes

2. Has the statistical analysis been performed appropriately and rigorously? 

Reviewer #1: Yes

Reviewer #2: Yes

3. Have the authors made all data underlying the findings in their manuscript fully available?

Reviewer #1: No

Reviewer #2: No

4. Is the manuscript presented in an intelligible fashion and written in standard English?

Reviewer #1: Yes

Reviewer #2: Yes

5. Review Comments to the Author

Reviewer #1: 1. Thanks for your research about the safety of elective PCI. The term "elective" is unusual as you have many NSTEMI and UA in your registry that require urgent or semi urgent PCI. For the other cases, i haven't found if the ischemia was previously documented and, when documented, the severity of ischemia.

Despite your propensity matched score, we observe more prior MI, more CVD, more previous HF, more left main disease in the group of patients who benefit from a PCI in a site with a surgical department.

2. Femoral access remains high 52% and maybe discussed

3. the same for the high use of Ticagrelor, high probably for NSTEMI. Therefore the term "elective" may be changed.

3. These data confirm several previous data. They are reassuring for your state, but what are the new informations for the readers ?

4. Multivessel disease and poylarterial disease are not reported and may be on interest in the discussion.

5. What's your definition of "high risk" patients and low risk patients ?

Thanks for your contribution

Reviewer #2: The authors have undertaken an analysis of the BMC-2 registry studying in hospital and post discharge outcomes in patients undergoing PCI startified by whether the PCI was undertaken in a surgical vs non-surgical centre. The authors report no differences in in-hospital outcomes, 90 day readmissions or cost of procedure following PSM analysis in the whole population aswell as a high risk subgroup. The analysis adds to the large body of literature published in this arena already, but presents additional data including the readmission data, and a subgroup analysis around a higher risk cohort. The manuscript is well written. i have number of recommendations:

1) Throughout the mnanuscript the authors refer to the cohort as an elective PCI cohort. This is misleading, UA and NSTEMI cases for the majority of PCI procedures in this analysis. They should probably rename their cohort as a nonPPCI or ACS / elective cohort.

2) Their choice of MACE is unusual. they have included CIN (which contributes the majority of outcome events). This is inappropriate. MACCE should probably be in hospital mortality, CVA, AMI/ Stent thrombosis and perhaps re-intervention. NACE could include major bleeding.

3) the 90 days data is limited to only 10% of population that they could match to medicare claims database. I think that this should be more overt in the abstract, as it is slightly misleading otherwise. the limitations should be re-emphasised in the limitations section of discussion

4) It would be good to have a figure showing centre volume in the individual centres, as i think that is useful for the reader. where there is a lot of data around centre-volume outcomes, do the authors think it is a good idea that activity is redistributed from higher to lower volume centres? (authors report redictribution of activity and discuss it in the context of patients having PCI more locally)

5) urgent / emergent surgery-was this due to complications? perhaps if data available re indiciation of this surgery would be very interesting as very little of this type of data.

Overall an interesting analysis that could be imporved with some additional work.

6. PLOS authors have the option to publish the peer review history of their article (what does this mean?). If published, this will include your full peer review and any attached files.

Reviewer #1: Yes: Pierre SABOURET

Reviewer #2: Yes: Mamas A. Mamas

---

## [Author Response · Author response to Decision Letter 0]

22 Jul 2020

See attached response to reviewers document for full responses

---

## [Decision Letter · Decision Letter 1]

10 Aug 2020

Trends and outcomes of non-primary PCI at sites without cardiac surgery on-site: The early Michigan experience

PONE-D-20-15844R1

Dear Dr. Gurm,

We’re pleased to inform you that your manuscript has been judged scientifically suitable for publication and will be formally accepted for publication once it meets all outstanding technical requirements.

Kind regards,

Giuseppe Andò, M.D., Ph.D.

Academic Editor

PLOS ONE

Additional Editor Comments (optional):

Reviewers' comments:

Reviewer's Responses to Questions

**Comments to the Author**

1. If the authors have adequately addressed your comments raised in a previous round of review and you feel that this manuscript is now acceptable for publication, you may indicate that here to bypass the “Comments to the Author” section, enter your conflict of interest statement in the “Confidential to Editor” section, and submit your "Accept" recommendation.

Reviewer #2: All comments have been addressed

2. Is the manuscript technically sound, and do the data support the conclusions?

Reviewer #2: Yes

3. Has the statistical analysis been performed appropriately and rigorously? 

Reviewer #2: Yes

4. Have the authors made all data underlying the findings in their manuscript fully available?

Reviewer #2: No

5. Is the manuscript presented in an intelligible fashion and written in standard English?

Reviewer #2: Yes

6. Review Comments to the Author

Reviewer #2: the authors have satisfactorily answered all my comments and addressed the issues that i raised. manuscript requires no further changes

7. PLOS authors have the option to publish the peer review history of their article (what does this mean?). If published, this will include your full peer review and any attached files.

Reviewer #2: No

---

## [Editor Report · Acceptance letter]

17 Aug 2020

PONE-D-20-15844R1 

Trends and outcomes of non-primary PCI at sites without cardiac surgery on-site: The early Michigan experience 

Dear Dr. Gurm:

I'm pleased to inform you that your manuscript has been deemed suitable for publication in PLOS ONE. Congratulations! Your manuscript is now with our production department. 

Kind regards, 

on behalf of

Dr. Giuseppe Andò 

Academic Editor

PLOS ONE